# Neuroelectric Correlates of Perceptual Awareness During the Auditory Attentional Blink

**DOI:** 10.3390/brainsci15060537

**Published:** 2025-05-22

**Authors:** Claude Alain, Mary O’Neil, Lori J. Bernstein, Dawei Shen, Bernhard Ross

**Affiliations:** 1Rotman Research Institute, Baycrest Academy for Research and Education, Toronto, ON M6A 2E1, Canada; moneil@research.baycrest.org (M.O.); shen_dawei@hotmail.com (D.S.);; 2Department of Psychology, University of Toronto, Toronto, ON M5S 3G3, Canada; 3Institute of Medical Sciences, University of Toronto, Toronto, ON M5S 3H2, Canada; 4Music and Health Science Research Collaboratory, University of Toronto, Toronto, ON M5S 1C5, Canada; 5Department of Psychiatry, University of Toronto, Toronto, ON M5S 3G3, Canada; lori.bernstein@uhn.ca; 6Department of Supportive Care, University Health Network, Toronto, ON M5G 2M9, Canada

**Keywords:** attentional blink, auditory, EEG, ERP, awareness-related negativity

## Abstract

Background: Perceptual awareness refers to the conscious detection and identification of a sensory event. In electrophysiological studies, it is associated with a modality-specific negative-going event-related potential, which can be observed as early as 100–300 ms after the stimulus onset. Method: In this study, we measured neuroelectric brain activity during the auditory attentional blink, comparing brain responses when participants correctly reported both the first (T1) and second (T2) targets versus when only T1 was detected, but T2 was missed. To achieve robust statistical power, we pooled data across six previously published studies for the current analyses. Result: Our results revealed that accurately reporting both T1 and T2 elicited greater negativity between 150 and 300 ms over the frontocentral and central scalp areas following T2 onset, compared to trials where T1 was detected but T2 was not. Additionally, a positive displacement, peaking around 800 ms over the central-parietal scalp area, followed the early negativity. Successful detection of both T1 and T2 was also associated with more pronounced alpha suppression, peaking at approximately 500 ms before and 800 ms after T2 onset. Conclusions: These findings suggest that neural correlates of what we refer to “auditory awareness” occur both before the stimulus sequence and soon after T2 onset. Pre-stimulus difference in alpha power may serve as an indicator of lapses in attention, with greater alpha power reflecting periods when participants are less engaged or off-task.

## 1. Introduction

Our auditory environment comprises many sounds, with only a subset being part of our perceptual experience. The latter is called perceptual awareness, which can be operationalized as the explicit reporting of hearing a particular stimulus [1]. Auditory perceptual awareness has been studied using bistable stimuli, such as repetitive tones or speech sounds that can be heard as a single stream or as two separate ones [1,2,3,4,5] or by comparing neural activity when a task-relevant stimulus is detected versus when it is not [6,7]. In the latter case, Gutschalk et al. [6] observed a negative wave between 70 and 300 ms post-stimulus onset when participants reported hearing the task-relevant stimulus (i.e., target) compared to when they did not. The amplitude distribution of this negative wave, referred to as “awareness-related negativity,” is consistent with neural generators located in the auditory cortex along the superior temporal gyrus. Notably, both detected and undetected targets generate comparable exogenous middle-latency evoked responses originating from the auditory cortex. This suggests that the awareness-related negativity is elicited after sensory registration and may index the conscious perception of a sound. However, the awareness-related negativity may also reflect fluctuations in endogenous attention when the incoming stimulus is compared to a representation of the designated target stimulus. In other words, awareness-related negativity could reflect the difference between trials in which the comparison process yields a “match” versus when it yields a “no match” signal.

The present study used an attentional blink paradigm to examine brain activity associated with perceptual awareness. In a typical attentional blink study, participants are presented with a stream of visual (e.g., letters) or auditory (e.g., tones) stimuli at a high rate (e.g., 8 or 10 per second). The attentional blink refers to a brief period—usually 200–500 milliseconds—after detecting a primary target (T1) during which a secondary target (T2) is less likely to be consciously reported. In our paradigm, participants listened to a rapid sequence of auditory stimuli (pure tones of different frequencies) that could comprise a first (T1) target (six 5-ms pulses) and a second (T2) target (tone glide). When both auditory targets are present and occur closer together in time, participants often fail to notice or report the second (i.e., T2) target, a phenomenon commonly referred to as the attentional blink. The attentional blink may reflect a deficit in consolidation, where the sensory trace elicited by T2 is not transferred into working memory [8,9,10].

The consolidation account is supported by event-related potential (ERP) studies showing reduced P3b amplitude in the attentional blink condition [11,12,13,14,15,16], an ERP component associated with context updating [17,18,19,20,21]. The P3b is sometimes preceded by an early negativity at central sites between 100 and 300 ms after T2 onset [11]. The latter shows a similarity with awareness-related negativity. These ERP modulations were revealed by subtracting ERPs elicited by T1 alone from ERPs when both T1 and T2 were presented. This difference wave minimizes ERPs elicited by the distractors and highlights neural activity associated with the attentional blink. However, this approach has weaknesses; namely, it uses conditions with different stimulus configurations, making it difficult to determine whether the difference wave isolated the awareness-related negativity or some other neural responses related to the stimulus configuration, such as an exogenous negative deflection peaking at about 100 ms after sound onset (i.e., the N1 wave) or a change detection negativity referred to as the mismatch negativity (MMN).

To address these concerns, Shen et al. [22] directly compared ERPs for detected versus undetected T2 targets in the attentional blink condition (i.e., when T2 closely followed T1). They found a larger P3b response to hits than misses (for a similar result for visual AB see, [23]), but no difference in earlier components preceding the P3b. This suggests that perceptual awareness may emerge only after the stimulus has been consolidated in working memory. However, only a subset of participants was included in that analysis, potentially limiting the statistical power to detect the awareness-related negativity.

In the present study, we re-analyzed EEG data from six auditory attentional blink experiments [11,22,24,25,26,27], each involving distinct participant samples. Focusing exclusively on the attentional blink condition, we categorized trials based on participants’ behavioral responses. Then, we compared neural activity between trials where both T1 and T2 were correctly identified and only T1 was detected. We hypothesized that correctly detecting both targets would be associated with an awareness-related negativity, relative to trials in which T2 was missed.

In our prior studies, the small or nonsignificant differences in ERP amplitude within the 100–300 ms interval could have been related to lapses of attention and episodes of mind wandering [11,22]. Prior studies have shown enhanced alpha power during lapses of attention or mind wandering [28,29,30,31,32,33]. Hence, in addition to time domain analysis, we conducted time-frequency analysis to assess the link between pre-stimulus alpha power and ERP amplitudes in trials where both T1 and T2 were correctly detected versus those where only T1 was detected. We anticipated that a greater alpha power would precede the presentation of T1 and T2, particularly in trials where T2 was missed.

## 2. Methods

### 2.1. Participants

Sixty-six participants (Mean age = 23.1 years; SD = 3.48; 39 female) who took part in one of six auditory attentional blink EEG studies conducted by our group [11,22,24,25,26,27] met the criteria for the present study. We pooled data across these studies to ensure sufficient statistical power for the present analyses. Participants were between 18 and 30 years of age, had normal hearing assessed using pure tone audiometry, and had no history of psychiatric, neurological, or other significant illnesses. In the sample of participants, performance at the attentional blink condition ranged from 52% to 85% accuracy (i.e., correctly detecting T2 in the trials where the T1 was also correctly detected). All participants had at least 15 trials included in the ERP average.

### 2.2. Stimuli and Procedure

In all studies, participants were presented with a sequence of 16 sounds. Each sound had a duration of 30 ms, including 2 ms linear onset and offset slopes. The stimulus onset asynchrony (SOA) was fixed at 90, 105, 120, or 150 ms, depending on the experimental condition and study. Standard sounds, termed distractors, were pure tones with their frequency randomly chosen out of 21 frequencies equally spaced on a logarithmic scale between 529 and 1330 Hz. The first target sound (T1) was composed of six identical 5 ms pure tones sliced together. The target frequency was randomly selected from the set of 21 frequencies. The second target sound (T2) was a fast frequency-modulated glide that changed smoothly from 636 to 1006 Hz. The stimuli were synthesized at a sampling rate of 44,100 Hz using Adobe Audition 1.5. They were presented binaurally at a 75 dB sound pressure level using insert earphones (ER-3A, Etymotic Research, Elk Grove, IL, USA). Stimulus presentation was controlled using Presentation software (version 13.0, Neurobehavioral Systems, Albany, CA, USA).

Figure 1 shows a schematic of a typical attentional blink condition in our studies. The first target (T1) was the fifth or seventh sound in the sequence. The second target (T2) was presented immediately after T1 [11,22,24] or at the second position (~200 ms) after T1 [25,26,27]. All studies included control conditions in which T2 was presented at the seventh or eighth position after T1. Participants were also presented with additional control conditions, where only T1 or T2 was presented, or neither T1 nor T2 were presented. After each stimulus sequence, participants indicated whether T1 and T2 were presented. The questions appeared in the center of the computer screen and remained until the participant responded by pressing “1” for “yes” or “2” for “no” on a keyboard.

The auditory stimulation and electroencephalography (EEG) recording took place in a double-walled, sound-attenuating booth. Participants sat one meter in front of a screen. They were instructed to keep their eyes open and to maintain their gaze on the fixation cross to minimize eye movements. A white fixation cross appeared in the center of the computer screen, starting 1000 ms prior to sequence onset and ending 500 ms after the sequence. Participant’s compliance with the instructions was monitored using a video camera.

Participants familiarized themselves with the stimuli and tasks in a practice block, during which all trial types were repeated twice. All participants met a criterion of at least 60% correct T1 and T2 detection when both were present in the sequence and T2 was presented at the +1 or +2 position.

### 2.3. EEG Recording

EEG was recorded initially from a 64- or 76-electrode montage using a NeuroScan SynAmps 2 (Compumedics, El Paso, TX, USA) or a BioSemi Active Two acquisition system (BioSemi V.O.F., Amsterdam, The Netherlands). All signals recorded using Neuroscan SynAmps 2 were bandpass filtered between 0.16 Hz and 100 Hz, whereas those recorded with BioSemi Active Two were bandpass filtered between 0.10 and 104.00 Hz. The continuous EEG was digitized at a 500 or 512 Hz sampling rate.

### 2.4. EEG Preprocessing

The data were analyzed using Brain Electrical Source Analysis (BESA) software (BESA Research 7.1; MEGIS, Gräfelfing, Germany). For each participant, ocular movements were measured at electrodes near the eyes, visually identified from the continuous EEG recording to generate a set of components that best explained the eye movement artifacts. The scalp projections of these components were then subtracted from the continuous EEG recording to minimize ocular contamination, such as blinks and lateral and vertical eye movements.

The EEG datasets sampled at 512 Hz were downsampled to a 500 Hz sampling rate. All EEG datasets were converted to a common 52-electrode montage shared by all partici-pants and comprised electrodes from the 10–20 system. The electrodes below and lateral to the eyes, FT9, FT10, TP9, TP10, P9, P10, and Iz were not included in the analyses, because they were missing in some participants and tend to be noisier than the scalp electrodes. The continuous EEG was bandpass filtered 0.5 to 90 Hz and parsed into epochs time-locked according to T2 onset, including a 1000 ms pre- and post-stimulus interval. The baseline correction used the mean amplitude between −1000 and −800 ms before T2 onset. Those epochs, including extreme values (peak-to-peak deflections exceeding 120 µV), were automatically marked and excluded from the averages. The proportion of epochs included in the averages ranged from 90 to 100% among the participants.

### 2.5. ERP Analysis

Our previous reports presented results related to attentional blink and how it varies as a function of various experimental conditions. The present study focused only on neural activity elicited during the attentional blink condition when both T1 and T2 were detected (Hit) versus when T1 was detected but T2 was missed. Only participants with ERP averages with more than 15 accepted trials for hits and misses were included in the ERP and time-frequency analysis. Preliminary analyses comparing the difference waves using clustered-based statistics and permutation tests (see below) did not reveal a significant interaction between condition (hits, misses) and T2 position (+1 vs. +2) nor SOAs. Therefore, the T2-locked evoked brain activity elicited by the +1 and +2 T2 position and SOAs were combined, and further analyses focused on the difference between hits and misses.

ERP source analysis BESA Research software was used to estimate distributed source activity associated with the awareness-related negativity. We modeled the group mean difference wave using an iterative application of Low-Resolution Electromagnetic Tomography (LORETA), which reduced the source space in each iteration. This imaging approach, termed Classical LORETA Analysis Recursively Applied (CLARA), yields more focal estimates of brain activity and can distinguish sources located in close proximity. The voxel size in Talairach space was 7 mm; this setting is appropriate for distributed images in most situations. The regularization parameters that account for the noise in the data were set with a single value decomposition cutoff at 0.01%. We used a four-shell ellipsoidal head model with a head radius of 85 mm, and thickness for scalp, bone, and CSF of 6, 7, and 1 mm, respectively. The relative conductivities were 0.33, 0.33, 0.0042, and 1 S/m for brain, scalp, bone, and CSF, respectively.

### 2.6. Time–Frequency Analysis

We employed a complex demodulation method with 1 Hz wide frequency bins and a 50 ms time resolution within the 2–50 Hz range to decompose single-trial EEG data into a time-frequency representation. To account for individual differences in baseline activity, the mean signal power for each participant was normalized to a baseline interval, which was defined as the period between −1000 and −800 ms before T2 onset. Our primary focus in the time-frequency analysis was on the alpha band (8–12 Hz) because this frequency range has been consistently linked to attention-related processes in previous studies [26,34]. By isolating this specific frequency band, we aimed to explore its role in perceptual awareness during the auditory attentional blink.

### 2.7. Statistical Analysis

The neural activities elicited by hits and misses were compared using cluster-based statistics and permutation tests (BESA Statistics 2.1). First, a series of t-tests were used to identify clusters in time (adjacent time points) and space (adjacent electrodes) where ERPs or oscillatory activity differed between the two conditions. The channel diameter was 4 cm, allowing for up to four electrode neighbors per analysis cluster. We used a cluster alpha of α = 0.05 for cluster building. A Monte-Carlo resampling technique [35] was then used to identify those clusters with higher values than 95% (one-sided *t*-test) of all clusters derived by random permutation of the data. The number of permutations was set at 5000. Importantly, this procedure corrected for multiple comparisons over time and electrodes to minimize false positives.

Finally, to examine the brain-behavior relationship, we used Spearman’s rho correlation analyses to assess the relationship between accuracy in the attentional blink condition, ERP amplitude, and pre-stimulus alpha power. For each participant, we extracted the amplitude of the awareness-related negativity and P3b from nine central-parietal electrodes (C1, Cz, C2, CP1, CPz, CP2, P1, Pz, P2). For pre-stimulus alpha power, we used the mean alpha power between −750 and −250 before T2 over the right central and central-parietal sites (C4, CP6, CP2, C6, C1, C2, Cz).

## 3. Results

Figure 2a shows the group mean ERPs elicited by hits and misses time-locked on T2 onset at the midline central-parietal site (CPz). The ERP amplitude before T2 onset was comparable for hits and misses. However, the ERPs associated with correctly identifying T1 and T2 were more negative between 100 and 300 ms after T2 onset than when participants failed to report the presence of T2. This early modulation, referred to as the awareness-related negativity, was followed by an enhanced positivity peaking at about 800 ms in the midline right central areas referred to as the P3b.

A cluster-based statistic for ERP amplitude revealed significant differences between the ERPs elicited by hits and misses during the attentional blink condition (Table 1). The first, third, and fourth clusters corresponded to the positive modulation with larger positivity for hits than misses at central and parietal sites, which reversed polarity at frontopolar and inferior frontal electrodes (Cluster #3 and #4). The second cluster showed a difference in ERP amplitude between hits and misses that peaked at about 236 ms after T2 onset over the central and central-parietal scalp areas.

Figure 2b shows the iso-contour maps for the difference in ERP amplitude between hits and misses. The contour maps depict the mean amplitude for the early awareness-related negativity (216–256 ms interval) and late P3b (780–820 ms interval) modulations following T2 onset. The difference in ERP amplitude between 216 and 256 ms after T2 onset was most prominent over the left central-parietal scalp area and inverted in polarity at mid-temporal and inferior temporal-parietal sites, consistent with generators located in the superior temporal gyrus. The difference in ERP amplitude between 780 and 820 ms after T2 onset was largest over the right central parietal area and inverted in polarity at frontopolar sites.

To better understand the neural sources underlying the observed difference in ERP amplitudes and to facilitate comparison with previous studies on perceptual awareness, we conducted a distributed source analysis using CLARA. Applying this technique to the group-averaged difference waveform revealed bilateral activation in the superior temporal gyrus during the interval associated with the awareness-related negativity (Figure 3), suggesting early auditory processing involvement in perceptual awareness. We also observed source activity in the middle temporal gyrus during the interval associated with the P3b. These sources were more posterior and inferior to the sources observed for the awareness-related negativity.

Lastly, to complement the ERP analysis and source-level findings, we examined the temporal dynamics of oscillatory activity using temporal spectral evolution analysis. Figure 4 shows the group mean time-frequency spectrograms for the entire trial, time-locked to T2 onset. Both hits and misses were associated with transient theta power (~5 Hz) at sequence onset, followed by sustained alpha power (~10 Hz) throughout the stimulus sequence, with the strongest activity observed over the left and right parietal scalp areas. The cluster analysis procedure and permutation-based statistics revealed two significant spatiotemporal clusters (Table 2). The first cluster emerged 550 ms after T2 and persisted until the end of the epoch. The second cluster reflected greater pre-stimulus alpha synchronization, beginning 750 ms before T2 onset and lasting until 250 ms before T2 onset.

### Analysis of Brain–Behavior Relationship

To further investigate the functional role of our EEG findings, we computed Spearman rho correlations between accuracy in the attentional blink condition, ERP amplitude, and pre-stimulus alpha power. We extracted the mean amplitude between 200 and 260 ms for the awareness-related negativity and between 750 and 850 ms for the P3b modulation from the difference wave at nine central-parietal electrodes for each participant. We also extracted the mean alpha power from the time–frequency analysis. We averaged the alpha power from the same four electrodes used in the previous analysis (C5, T7, TP7, CP5), regardless of the SNR conditions.

We observed a significant correlation between the magnitude of the awareness-related negativity and P3b amplitude (rs = −0.271, *p* = 0.028), indicating that a more pronounced negativity between 200 and 260 ms was associated with more pronounced positivity between 750 and 850 ms following T2. However, correlations between the attentional blink performance and the awareness-related negativity (rs = −0.101, ns) or P3b (rs = 0.012, ns) amplitude were not significant, nor were the correlations between pre-stimulus alpha power and either ERP component (rs < −0.118, in both cases).

## 4. Discussion

The correct detection of T1 and T2 was associated with an awareness-related negativity peaking at about 230 ms following T2 onset, followed by greater P3b amplitude. The timing and scalp amplitude distribution of this awareness-related negativity was similar to those reported in previous studies using a pre-defined target embedded in distractors [1,6,36,37]. In the present study, the awareness-related negativity was largest over central scalp sites and showed inverted polarity at inferior parietal-occipital sites, consistent with sources located in bilateral superior temporal gyri. The findings from our distributed source analysis provide further support for awareness-related activity localized in auditory areas when participants perceived T2. This aligns with findings from magnetoencephalography showing awareness-related negativity sources in the superior temporal gyrus [6] and with an intracranial EEG study demonstrating stronger gamma-evoked activity from the auditory cortex of patients with epilepsy when they accurately detected a near threshold auditory stimulus [38]. Moreover, using high-field fMRI at 7 Tesla, Heynckes et al. [39] reported a difference in neural activity within primary auditory areas for detected versus undetected targets, with no significant differences in neural activities in the planum polare or planum temporale. Lastly, our results are also consistent with visual attentional blink studies reporting enhanced early negativities over occipital and inferior-parietal scalp areas for detected versus undetected stimuli in the attentional blink condition [40,41,42,43]. Together, these findings reveal a strong association between neural activity in sensory areas and perceptual awareness. The findings from the present study differ from those of a prior study from our group focusing on the attentional blink condition, which only reported greater P3b amplitude when T1 and T2 were both detected [22]. Here, leveraging a sample size three times larger than in our prior study, we observed a clear and unequivocal awareness-related negativity associated with the participants’ perception of T2. Our findings highlight the importance of using a large sample size when investigating the neural correlates of perceptual awareness.

In addition to perceptual awareness, the difference in T2-evoked-related activity between hits and misses observed in the present study could reflect attentional allocation, with greater attention-related negativity when participants correctly detected both targets than when only the first target was correctly identified. Prior studies of auditory selective attention have shown greater negativity between 100 and 300 ms after sound onset for attended (i.e., task-relevant) than unattended (i.e., task-irrelevant stimuli) [44,45,46,47,48]. Attention-related negativity is also greater in amplitude when the task-relevant stimuli belong to a perceptual auditory object [49,50]. In the context of the temporal attention blink paradigm, hit trials indicate more focused attention, which is related to greater attention-related negativity. In contrast, weaker attentional focus or lapses of attention lead to trials where the second target T2 is undetected.

In the present study, oscillatory brain activity observed before T2 exhibited a distinct 10 Hz rhythm, which aligns with the stimulation rate used in our attentional blink experiments. Interestingly, this 10 Hz entrainment was evident despite slight variations in the presentation rate across studies. After T2, however, the 10 Hz entrainment disappeared, even though the stimulus sequence continued. Previous research suggests that alpha power is a key indicator of attentional engagement, with a lower alpha power reflecting focused attention on the task, while a higher alpha activity is associated with attention lapses or mind-wandering [28,29]. If participants’ attention drifts during the rapid serial auditory presentation (RSAP) attention task, this could hinder T2 detection. The increased alpha power observed prior to T2 onset in our study supports this hypothesis.

The awareness-related negativity observed in the present study was followed by a positive displacement peaking at about 800 ms after T2 onset over the parietal region. The positive displacement bears similarities to the P3b component, although it exhibited a more lateralized and anterior distribution than the typical P3b. This modulation may reflect post-perceptual processes associated with decision-making and response preparation rather than awareness. The delayed and/or reduced P3b in the attentional blink condition has often been interpreted as indicative of a deficit in memory consolidation [11,12,13,51]. However, our findings reveal a difference in pre-stimulus alpha power, consistent with a lapse of attention. Thus, the deficits in memory consolidation may be related to the fact that the participants’ attention was not fully engaged during that trial.

### Implications and Limitations

This study reveals a clear awareness-related negativity during the temporal attention blink paradigm. These findings have practical applications: for instance, awareness-related negativity may serve as a neural marker to evaluate the effectiveness of cognitive training paradigms designed to enhance attention deployment and maintenance in complex auditory environments. Such interventions could help individuals better attend to and process acoustic information that might otherwise be overlooked or misinterpreted, thereby enhancing both automatic auditory processing and conscious awareness of an auditory scene.

While comparisons to attention-related ERP components such as the N2 or MMN could help clarify the distinction between perceptual awareness and attentional processes, our paradigm does not reliably elicit these components, limiting such comparisons in the current dataset. Future studies designed to concurrently elicit awareness-related and attention-related ERPs could help dissociate these overlapping but distinct neural processes.

Our study examined awareness-related negativity during an auditory temporal attention blink paradigm using pure tone distractors. This methodological choice limits the generalization of the findings to paradigms involving similar, controlled experimental contexts. To establish broader applicability, future research should explore awareness-related negativity in more ecologically valid settings with complex or naturalistic auditory stimuli.

We found no significant correlations between attentional blink performance and either pre-stimulus alpha power or the amplitude of awareness-related negativity and P3b components. This lack of associations may be attributable to our experimental design, in which participants responded to T1 and T2 only at the end of each sequence, rather than immediately upon perceiving the second target (T2). Additionally, our approach to isolating the awareness-related negativity—contrasting hits and misses within the same stimulus condition—likely reduced the P3b amplitude, in contrast to other studies that contrasted attentional blink trials with control trials lacking a T2 stimulus [11,22,24,25,26,27]. Future studies could incorporate probability-based timing manipulations to examine how temporal expectations influence pre-stimulus alpha power and P3b amplitude, shedding light on the role of temporal attention in auditory awareness.

While pooling data across studies increases statistical power, it may also introduce variability due to methodological differences, such as varying temporal intervals between T1 and T2. These variations can differentially influence the overlap of neural responses to T1 and T2, complicating the interpretation of hit-versus-miss comparisons. Differences in experimental design and data acquisition protocols may also obscure relationships between behavioral performance and neural measures. Nevertheless, the consistency of the awareness-related negativity across multiple datasets—despite differences in recording systems and methodologies—underscores the robustness and reliability of this neural marker.

## 5. Conclusions

This study demonstrates that successful identification of both T1 and T2 is associated with three distinct neural signatures: pre-stimulus alpha suppression, an awareness-related negativity peaking around 230 ms after T2, and a later P3b peaking around 800 ms. Reduced pre-stimulus alpha power appears to reflect heightened attentional readiness, whereas elevated alpha power may indicate lapses of attention in trials where T2 is not consciously perceived. The amplitude distribution of the awareness-related negativity is consistent with generators located in the auditory cortex along the superior temporal gyrus, while the later P3b aligns with cognitive processes such as decision-making and response preparation. Together, these findings reveal a temporal cascade of neural activity supporting conscious perception and subsequent cognitive evaluation. Future studies employing intracerebral recordings in epilepsy patients or high-field fMRI could help localize the sources of these signals with greater precision, offering deeper insight into the neural mechanisms underlying attention, awareness, and decision-making.

## Figures and Tables

**Figure 1 brainsci-15-00537-f001:**
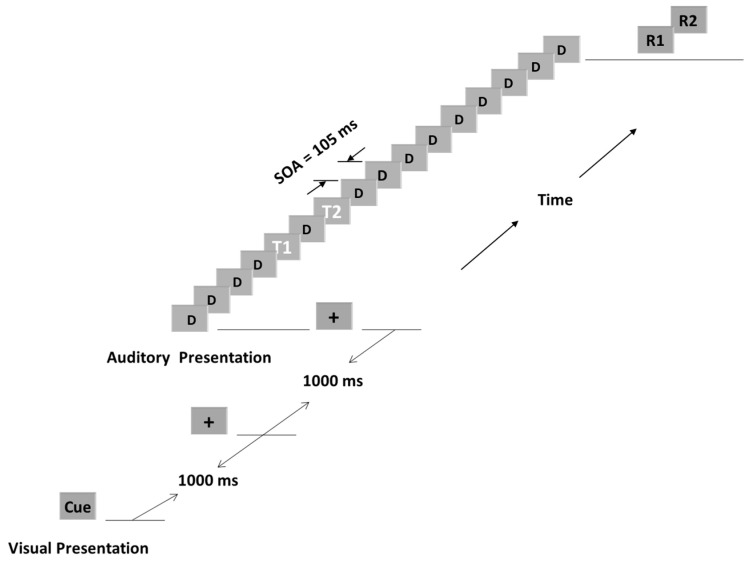
*Schematic of a single trial in a typical attentional blink paradigm*. Each trial began with a fixation cross displayed on a computer screen, followed by a sequence of 16 auditory stimuli. In the illustrated example, the first auditory target (T1) occurs at the 5th position, with the second target (T2) occurring two positions later. Distractors (D) are non-target stimuli interspersed throughout the sequence. R1 and R2 represent responses to T1 and T2, respectively. The stimulus onset asynchrony (SOA) denotes the time interval between the onsets of any two stimuli.

**Figure 2 brainsci-15-00537-f002:**
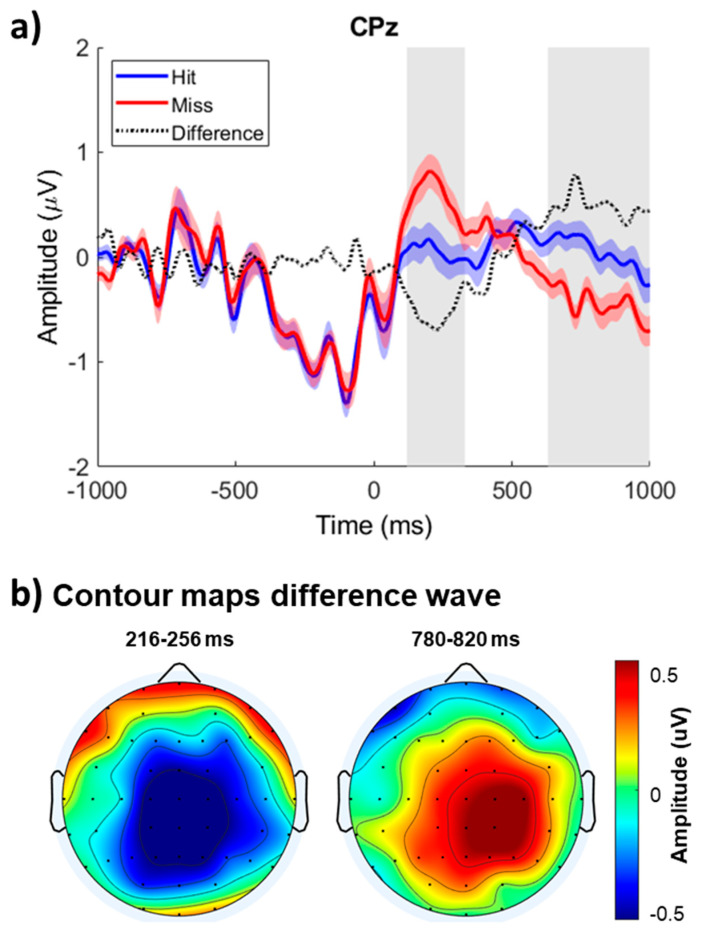
Brain responses elicited when the first (T1) and second (T2) targets were correctly detected (Hits) versus when only T1 was detected (Misses). (**a**) Group mean event-related potential time-locked on T2 onset (0 ms) at the midline central parietal electrode (CPz) for hits and misses. The color shaded areas show the group mean standard deviation. The gray shaded areas show the interval where the amplitude difference was significant. (**b**) Iso-contour maps for the difference in ERP amplitude between hits and misses. The contour maps show the mean amplitude for the early (216–256 ms) and late (780–820 ms) modulations following T2 onset.

**Figure 3 brainsci-15-00537-f003:**
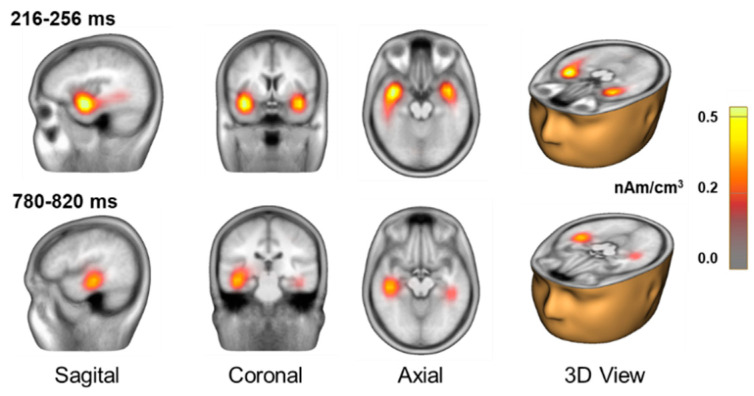
*Brain electrical source analysis.* Distributed source analysis of group mean difference wave using an iterative application of Low-Resolution Electromagnetic Tomography (LORETA) termed Classical LORETA Analysis Recursively Applied (CLARA). The top panel shows the mean source activity for the 216–256 interval, whereas the bottom panel shows the solution for the 780–820 ms interval.

**Figure 4 brainsci-15-00537-f004:**
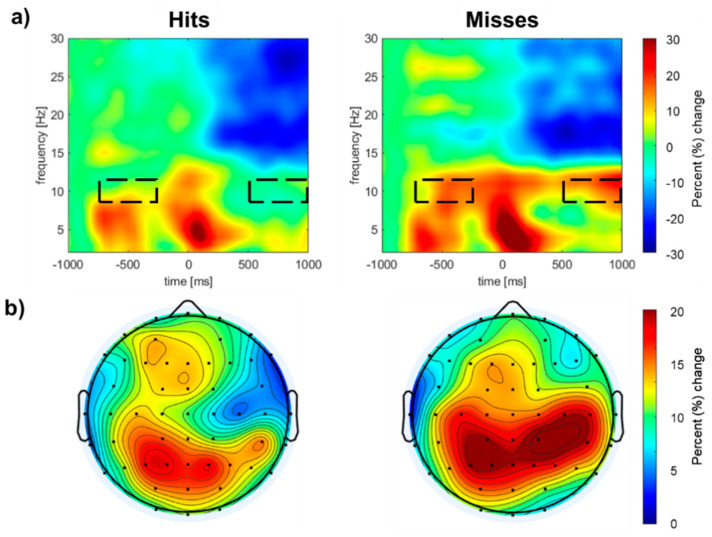
*Time–frequency analysis.* (**a**) Alpha power (% change relative to baseline −1000 to −800 ms) before and after the onset of the secondary target. Time–frequency for hits (**left panel**) and misses (**right panel**) from the right central electrode (C4). The dashed box highlights the intervals of differences in alpha power (8–12 Hz). Alpha synchronization was higher for the misses. (**b**) Contour map of pre-stimulus mean alpha power for the −550 to −450 ms interval for hits and misses.

**Table 1 brainsci-15-00537-t001:** Clustered statistics of ERP amplitude.

Cluster #	Range (ms)	Peak Latency/Electrode	Electrodes	*p* Values
1	616 to 998	798 ms/C4	C4, P4, P3, O2, O1, P7, Pz, TP7, Oz, PO4, PO3, CP5, CP6, CP1, CP2, FC2, FC1, CPz, P1, POz, P2, P6, P5, C1, C2, F2, FCz, Cz	<0.001
2	92 to 430	236 ms/CP1	F4, C3, C4, P4, P3, P8, Pz, TP7, TP8, PO4, PO3, CP5, CP6, CP1, CP2, FC2, FC1, FC6, CPz, P1, POz, P2, P6, C6, P5, C1, C2, C5, F2, F1, FCz, Cz	<0.001
3	526 to 704	646 ms/FPz	Fp1, Fp2, F4, F8, F7, AF3, AF4, F6, AF8, F5, AF7, Fpz	0.004
4	706 to 998	740 ms/AF7	Fp1, Fp2, F4, F8, F7, AF3, AF4, F6, AF8, F5, AF7, Fpz	0.006

**Table 2 brainsci-15-00537-t002:** Clustered statistics of alpha power (8–12 Hz).

Cluster #	Range (ms)	Peak Latency/Electrode	Electrodes	*p* Values
1	550 to1000	850/C4	Fp1, Fp2, F4, F3, C3, C4, P4, P3, O2, O1, F8, F7, T8, T7, P8, P7, Pz, Fz, TP7, TP8, Oz, PO4, PO3, CP5, CP6, CP1, CP2, FC2, FC1, AF3, AF4, FC6, FC5, CPz, P1, POz, P2, P6, C6, P5, C1, C2, C5, F2, F6, F1, AF8, F5, AF7, Fpz, FCz, Cz	<0.001
2	−700 to −250	−500/C4	C4, CP6, CP2, C6, C1, C2, Cz	=0.023

## Data Availability

The data presented in this study are available on request from the corresponding author due to (specify the reason for the restriction).

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
