# Peer review of "Neuroelectric Correlates of Perceptual Awareness During the Auditory Attentional Blink"

_brainsci, 2025, doi:10.3390/brainsci15060537_

Round 1
Reviewer 1 Report
Comments and Suggestions for Authors
This manuscript investigates the auditory attentional blink (AB) using a well-designed experimental paradigm, ensuring robust statistical power by pooling data from six studies. The EEG preprocessing and analysis follow standard, rigorous procedures, strengthening the reliability of the findings. The results are clearly presented, with detailed descriptions of event-related potential (ERP) components and time-frequency analyses.
The study reinforces the role of awareness-related negativity (ARN) in auditory perception and offers a novel perspective by linking alpha power fluctuations to attentional lapses. This connection provides fresh insights into the mechanisms of attentional engagement.
Areas for Improvement:
The introduction is dense and could be more accessible, especially for non-experts. Simplifying key concepts and providing additional background information would enhance readability.
More details on cluster-based statistics would help clarify how significant effects were determined.
Reporting effect sizes alongside p-values for key findings would strengthen statistical transparency.
A follow-up analysis could investigate whether temporal cues improve participant performance, revealing whether alpha oscillations are proactively adjusted rather than merely reflecting post hoc attentional lapses.
Source localization techniques could provide greater spatial precision, confirming whether these ERP markers originate from the same cortical networks identified in previous research.
A deeper analysis of the relationship between pre-stimulus alpha power and P3b amplitude could clarify whether attentional lapses predict weaker memory consolidation.
Comparing ARN to known attention-related ERPs (e.g., N2, MMN) could help differentiate perceptual awareness from general attentional processes.
Introducing probability-based timing manipulations for the probe could assess whether temporal expectation modulates alpha power and P3b amplitude, providing further insights into temporal attention mechanisms.
Comparing explicit temporal cueing with implicit temporal expectation in relation to alpha power and ERP components could offer a comprehensive understanding of how different forms of temporal attention influence AB.
Minor Concerns:
"Power" is missing after "alpha" in some sentences.
"wandering" in "mind-wandering wandering" should be removed (L256).
"Epileptic patients" should be revised to "patients with epilepsy" for more appropriate phrasing.
Author Response
We thank the reviewers for their time and constructive comments, which were very helpful in revising the manuscript. We have thoroughly revised our manuscript to address all the issues raised. All changes are detailed, point by point, in our response to the reviewers. The manuscript modifications significantly improve the original submission and contribute to the scientific community.
Comment 1: The introduction is dense and could be more accessible, especially for non-experts. Simplifying key concepts and providing additional background information would enhance readability.
Response 1: We revised several paragraphs of the introduction to improve clarity.
Comment 2: More details on cluster-based statistics would help clarify how significant effects were determined.
Response 2: In the method section, we added a paragraph describing the cluster-based statistic and permutation test.
Comment 3: Reporting effect sizes alongside p-values for key findings would strengthen statistical transparency.
Response 3: While reporting effect sizes alongside p-values is often considered good practice when using general linear models, this convention is not straightforward or meaningful when using cluster-based permutation tests. These methods are inherently non-parametric and designed to address multiple comparison issues in complex data structures, such as electrophysiological data, where assumptions of normality and independence are often violated. Importantly, the effect size interpretation is ambiguous in cluster-based inference. In cluster-based permutation tests, the primary unit of inference is the cluster, not the individual data point. Clusters are identified based on a thresholded statistic (e.g., t-value), and their cluster significance is assessed through permutation-derived distributions of cluster-level statistics (e.g., cluster mass or size). Because clusters vary in size, duration, and topographical extent, computing a single "effect size" for an entire cluster often oversimplifies and potentially misrepresents the spatial-temporal dynamics of the effect. Because of these reasons, we opt not to report effect sizes.
Comment 4: A follow-up analysis could investigate whether temporal cues improve participant performance, revealing whether alpha oscillations are proactively adjusted rather than merely reflecting post hoc attentional lapses.
Response 4: We thank the reviewer for this suggestion. While examining pre-stimulus alpha power as a function of cue type may shed some light on the hypothesis of lapse of attention, the number of observations for the hits and misses as a function of the cue condition is insufficient for reliable statistical analyses. However, we plan to expand on our prior published work on the effect of temporal cue (implicit and explicit) on accuracy, which has been reported in some earlier publications (Shen & Alain, 2011, 2012; Shen, Ross, & Alain, 2016). In those studies, we also report the effect of temporal attention on ERP and oscillatory activity.
Comment 5: Source localization techniques could provide greater spatial precision, confirming whether these ERP markers originate from the same cortical networks identified in previous research.
Response 5: We agree this is a good idea, and we did as suggested. We used distributed source analysis to examine the neural generator of the early negativity found. This new analysis reveals bilateral sources in the superior temporal gyrus. These findings and a figure showing the sources' location have been added into the results section.
Comment 6: A deeper analysis of the relationship between pre-stimulus alpha power and P3b amplitude could clarify whether attentional lapses predict weaker memory consolidation.
Response 6: We revised as suggested, and now report the results from the correlation analyses between pre-stimulus alpha power and P3b amplitude.
Comment 7: Comparing ARN to known attention-related ERPs (e.g., N2, MMN) could help differentiate perceptual awareness from general attentional processes.
Response 7: We appreciate the suggestion to compare the ARN with attention-related ERPs such as the N2 and MMN. However, the experimental design is not ideal for examining the difference between N2, MMN and ARN because the paradigm used in the present study does not generate a clear MMN or N2 wave, making it challenging to compare these ERP components. While we recognize the value of understanding how awareness-related ERPs like the ARN relate to attentional components, direct comparisons in the present experimental context would be methodologically and conceptually ambiguous. We added a sentence in the discussion about this in the Implications/Limitations section.
Comment 8: Introducing probability-based timing manipulations for the probe could assess whether temporal expectation modulates alpha power and P3b amplitude, providing further insights into temporal attention mechanisms.
Response 8: We thank the reviewer for this suggestion and will consider adding probability-based timing manipulations in future studies on temporal attention mechanisms. We added a sentence on this in the Implications/Limitations section.
Comment 9: Comparing explicit temporal cueing with implicit temporal expectation in relation to alpha power and ERP components could offer a comprehensive understanding of how different forms of temporal attention influence AB.
Response 9: We agree. In future studies, we will consider comparing implicit and explicit cueing effects on pre-stimulus alpha and neural signatures of the attentional blink.
Minor Concerns:
"Power" is missing after "alpha" in some sentences.
This has been corrected.
"wandering" in "mind-wandering wandering" should be removed (L256).
This has been corrected.
"Epileptic patients" should be revised to "patients with epilepsy" for more appropriate phrasing.
Thank you for pointing that out. This has been corrected.
Reviewer 2 Report
Comments and Suggestions for Authors
Major issues/remarks
The paper could be enhanced by clarifying certain concepts and explanations in a more elaborate way. I am especially referring to the figures in this case, but also a few methodological concepts were glanced over pretty quickly.
- Firstly, in the methods section I would consider adding a figure to illustrate the experimental design, this would make the manuscript easier to follow. For example a figure such as the one shown in Shen et al., (2018) (ref [10]) would be an addition to explain the paradigm to the reader who is less familiar with the paradigm.
- Figure 1 is pretty concise and it is not always clear from the methods how it was exactly obtained. In figure 1a, how are the group mean ERP and the significant grey areas obtained? In figure 1b, the contour maps aren’t specifically mentioned in the methods or results. Also the subtitles “ARN” and “P3” are in my opinion a bit confusing since these terms are not used in the results section describing the figure.
- Furthermore, the figures (both figure 1 and 2) currently present only the group mean ERP and group mean TSE. To gain a more comprehensive understanding of the findings, it could be valuable to examine the intersubject variability. Given the sample size of 66 participants, individual differences are bound to exist and could offer important insights into the variability of neural responses associated with perceptual awareness. Analysing these differences could enhance the robustness and applicability of the results.
- The statistical analysis isn’t explained in the methods section. Both the cluster-based statistic and Spearman’s rho are only introduced in the results section but are not really elaborated on. Especially the section about Spearman’s rho is very briefly mentioned in the results, whereas I would like some extra explanation here, mainly on why there is no correlation between the AB performance and the amplitude of the pos, neg modulations.
There are some possible changes that can be made to the discussion to enhance the manuscript.
- I would recommend the authors to expand the discussion on the generalizability of their results. It could prove useful to expand the discussion to include how these findings might relate to perceptual awareness in other sensory modalities, such as vision or touch. This could provide a more comprehensive understanding of the neural mechanisms underlying perceptual awareness.
- It could be interesting to mention the practical applications of these findings. For example, how the findings could inform cognitive training programs aimed at enhancing attentional focus and perceptual awareness.
- Discuss the limitation in more detail. Explore the potential impact of individual differences in attentional control and perceptual awareness. Discuss how these differences might influence the results and suggest ways to account for them in future research (related also to comment 1c) . As for the statistical power, while pooling data enhances statistical power, acknowledge the variability introduced by combining data from multiple studies. Discuss how this variability might affect the consistency of the findings and the ability to draw definitive conclusions.
With regard to the conclusion, make sure to provide a concise summary of the key findings in the conclusion, highlighting the most important insights gained from the study. Now it seems that the results concerning the alpha power are omitted? Is there a specific reason for this? Because it is mentioned in the abstract and in the results section.
Minor issues/remarks
Abstract
Line 25: ‘an’ should be ‘and’
Introduction
Line 49: T2 is defined as a second target but in line 57 it is referred to as ‘probe’. It might be better to already introduce this term here.
Line 56-58: This ‘early negativity at central sites between 100-300 ms after probe onset’ is supposed to show a similarity with the awareness related negativity. So the two are not the same and if so, why not?
Line 67-73: is it correct of me to interpret that this study builds on the results in Shen et al. [10]? If so, this was not entirely clear for me during the first time I read the manuscript, it could be useful to state this more clearly.
Line 80-81: to which small or non-significant awareness related negativity is this a reference? Is this referring to the study mentioned earlier by Shen et al. [10]?
Methods
Line 100-101: define the acronym for stimulus onset asynchrony here. Further in the methods section this acronym is used but it is never defined.
Line 103-104: what is the reason for this specific frequency range?
Line 104-106: Define again the target sound as T1 and the probe sound as T2, because now there are different terminologies being used. Also, if relevant, it might be interesting for the reader to add a figure showing T1 and T2 (the waveform).
Line 111: it is stated that the target is the fifth sound in the sequence, however for example in Shen et al. [10] the target is the seventh sound.
Line 137-138: how exactly were the ocular movements identified? Via EOG, manually, ICA,…?
Line 142: ‘decimated’ à ‘downsampled’
Line 158-159: Do I understand it correctly that first the ERP was calculated for each participant with a minimum of 15 trials per participant, and then later these individual ERPs are averaged to obtain the group mean ERP shown in figure 1? This was not entirely clear for me.
Line 160: how was it determined that there is no significant interaction between condition and T2 position, nor SOAs?
Results
Line 175: figure 1 is mentioned here, but actually only figure 1a is properly discussed
Line 175-176: see major comments, first comment
Line 179: Am I correct in assuming that the authors are referring to figure 1b when they mention the difference in ERP amplitude? Or is this still something different? If so, it would be good if figure 1b was properly mentioned/discussed.
Line 182-183: Same comment as previous, also a reference to figure 1b?
Line 192-193: see major comments, first comment
Line 193-198: Is it correct that this second cluster then represents then the negative difference in ERPs between 100-300 ms associated with correct detection of T1 and T2? And the other three clusters represent the enhanced positivity for this condition peaking at about 800 ms? This could be clarified a bit more.
Line 200-201: see major comments, first comment
Line 214-220: see major comments, first comment
Discussion & Conclusion section: see major comments, comments 2-5
Line 241: the term attention related negativity is introduced, is this different from the awareness related negativity? If no, it would be better to be consistent with the terminology.
Line 250-251: “… which aligns with the stimulation rate used in our AB experiments”. Is this related to the SOAs? “Interestingly, this 10-251 Hz entrainment was evident despite slight variations in the presentation rates across studies” Is there a more elaborate explanation for this? Why does this happen?
Line 256: 2x “wandering”
Line 259: Now the term awareness related negativity is used again, whereas in lines 224-226 it is stated that “The timing and amplitude distribution of the early negativity show similarities with the awareness-related negativity observed when listeners detect a pre-defined target embedded in distractors”. For me it is a bit confusing that this term is used only sometimes whereas in other parts of the manuscript the term is avoided.
Line 273: why is the acronym ARN only introduced now after writing the term in full during the whole paper?
Author Response
Comment 1: The paper could be enhanced by clarifying certain concepts and explanations in a more elaborate way. I am especially referring to the figures in this case, but also a few methodological concepts were glanced over pretty quickly.
Response 1: We have revised all sections of the manuscripts to enhance readability, with special attention to the method section.
Comment 2: Firstly, in the methods section I would consider adding a figure to illustrate the experimental design, this would make the manuscript easier to follow. For example a figure such as the one shown in Shen et al., (2018) (ref [10]) would be an addition to explain the paradigm to the reader who is less familiar with the paradigm.
Response 2: We thank the reviewer for this suggestion. We created a schematic of the paradigm and added it in the method section.
Comment 3: Figure 1 is pretty concise and it is not always clear from the methods how it was exactly obtained. In figure 1a, how are the group mean ERP and the significant grey areas obtained? In figure 1b, the contour maps aren’t specifically mentioned in the methods or results. Also the subtitles “ARN” and “P3” are in my opinion a bit confusing since these terms are not used in the results section describing the figure.
Response 3: We have revised the method section significantly. The revised manuscript expands on how ERPs were obtained. In addition, we added a paragraph on the cluster-based statistic and permutation test, which were used to test for differences between the ERPs elicited by hits and misses. The gray shaded areas show the interval where the amplitude difference was significant. This is now stated in the figure caption. We also revised the results section, where we discuss the contour maps. Lastly, we removed the ARN and P3 labels in what is now Figure 2 to avoid confusion.
Comment 4: Furthermore, the figures (both figure 1 and 2) currently present only the group mean ERP and group mean TSE. To gain a more comprehensive understanding of the findings, it could be valuable to examine the intersubject variability. Given the sample size of 66 participants, individual differences are bound to exist and could offer important insights into the variability of neural responses associated with perceptual awareness. Analysing these differences could enhance the robustness and applicability of the results.
Response 4: We appreciate the reviewer’s interest in intersubject variability. In the present study, we present the standard deviations for the ERPs, which helps illustrate variability.
Comment 5: The statistical analysis isn’t explained in the methods section. Both the cluster-based statistic and Spearman’s rho are only introduced in the results section but are not really elaborated on. Especially the section about Spearman’s rho is very briefly mentioned in the results, whereas I would like some extra explanation here, mainly on why there is no correlation between the AB performance and the amplitude of the pos, neg modulations.
Response 5: In the method section, we added a paragraph describing the cluster-based statistic and the permutation test. We also revised the results section on correlations and added a section on study limitations, explaining the lack of significant correlations.
Comment 6: I would recommend the authors to expand the discussion on the generalizability of their results. It could prove useful to expand the discussion to include how these findings might relate to perceptual awareness in other sensory modalities, such as vision or touch. This could provide a more comprehensive understanding of the neural mechanisms underlying perceptual awareness.
Response 6: We expanded the discussion on the generalizability of our results to the visual modality.
Comment 7: It could be interesting to mention the practical applications of these findings. For example, how the findings could inform cognitive training programs aimed at enhancing attentional focus and perceptual awareness.
Response 7: We added a few sentences about possible applications for cognitive training programs.
Comment 8: Discuss the limitation in more detail. Explore the potential impact of individual differences in attentional control and perceptual awareness. Discuss how these differences might influence the results and suggest ways to account for them in future research (related also to comment 1c) . As for the statistical power, while pooling data enhances statistical power, acknowledge the variability introduced by combining data from multiple studies. Discuss how this variability might affect the consistency of the findings and the ability to draw definitive conclusions.
Response 8: The revised manuscript acknowledge that while pooling data enhances statistical power, combining findings from different studies may introduce variability. The combinations of multiple studies demonstrate the robustness of the findings, which suggests that the awareness-related negativity is little affected by minor methodology differences between studies.
Comment 9: With regard to the conclusion, make sure to provide a concise summary of the key findings in the conclusion, highlighting the most important insights gained from the study. Now it seems that the results concerning the alpha power are omitted? Is there a specific reason for this? Because it is mentioned in the abstract and in the results section.
Response 9: We have revised the conclusion as suggested.
Minor issues/remarks
Abstract
Line 25: ‘an’ should be ‘and’
This has been corrected.
Introduction
Line 49: T2 is defined as a second target but in line 57 it is referred to as ‘probe’. It might be better to already introduce this term here.
We have replaced the term “probe” with “T2” to avoid confusion.
Line 56-58: This ‘early negativity at central sites between 100-300 ms after probe onset’ is supposed to show a similarity with the awareness related negativity. So the two are not the same and if so, why not?
The reference to probe onset has been replaced with T2 throughout the manuscript to avoid confusion.
Line 67-73: is it correct of me to interpret that this study builds on the results in Shen et al. [10]? If so, this was not entirely clear for me during the first time I read the manuscript, it could be useful to state this more clearly.
This study is motivated by prior studies on perceptual awareness, including Shen et al. We revised the introduction to make this more transparent.
Line 80-81: to which small or non-significant awareness related negativity is this a reference? Is this referring to the study mentioned earlier by Shen et al. [10]?
Thank you for pointing that out. We now provide the references.
Methods
Line 100-101: define the acronym for stimulus onset asynchrony here. Further in the methods section this acronym is used but it is never defined.
The acronym for stimulus onset asynchrony is now defined.
Line 103-104: what is the reason for this specific frequency range?
We used an extensive range of frequencies to minimize specific-frequency adaptation, which could yield pop-out effect.
Line 104-106: Define again the target sound as T1 and the probe sound as T2, because now there are different terminologies being used. Also, if relevant, it might be interesting for the reader to add a figure showing T1 and T2 (the waveform).
Thank you for pointing this out. We are now using T1 and T2 throughout the manuscript.
Line 111: it is stated that the target is the fifth sound in the sequence, however for example in Shen et al. [10] the target is the seventh sound.
Thank you for pointing that out. This was an oversight on our part. It is now corrected in the method section.
Line 137-138: how exactly were the ocular movements identified? Via EOG, manually, ICA,…?
From the continuous EEG, we visually identified ocular movements measured at electrodes near the eyes of each participant. This has been clarified in the method section.
Line 142: ‘decimated’ à ‘downsampled’
Thank you for the suggestions. We replaced “decimated” with “downsampled.”
Line 158-159: Do I understand it correctly that first the ERP was calculated for each participant with a minimum of 15 trials per participant, and then later these individual ERPs are averaged to obtain the group mean ERP shown in figure 1? This was not entirely clear for me.
Yes, this is correct. We have revised the method section and the figure caption to clarify this.
Line 160: how was it determined that there is no significant interaction between condition and T2 position, nor SOAs?
We computed the difference waves between hits and misses for each experiment and compared them between the experiments to ensure that they were comparable. The method section clarifies this.
Results
Line 175: figure 1 is mentioned here, but actually only figure 1a is properly discussed.
We have revised the results section, which now mentions the awareness-related negativity's scalp distribution.
Line 175-176: see major comments, first comment
We have revised the results section and provided more information about the pre-processing of EEG and statistical analyses.
Line 179: Am I correct in assuming that the authors are referring to figure 1b when they mention the difference in ERP amplitude? Or is this still something different? If so, it would be good if figure 1b was properly mentioned/discussed.
We have revised all sections of the manuscript to minimize confusion. After defining it in the first paragraph of the results section, we are using the term awareness-related negativity throughout the manuscript.
Line 182-183: Same comment as previous, also a reference to figure 1b?
The result section has been revised extensively.
Line 192-193: see major comments, first comment
All sections of the manuscript have been revised substantially to improve readability.
Line 193-198: Is it correct that this second cluster then represents then the negative difference in ERPs between 100-300 ms associated with correct detection of T1 and T2? And the other three clusters represent the enhanced positivity for this condition peaking at about 800 ms? This could be clarified a bit more.
This is correct. We have revised the result section to make it more straightforward.
Line 200-201: see major comments, first comment
We provide more information in each figure.
Line 214-220: see major comments, first comment
We have revised the paragraph on time-frequency analysis.
Discussion & Conclusion section: see major comments, comments 2-5
Line 241: the term attention related negativity is introduced, is this different from the awareness related negativity? If no, it would be better to be consistent with the terminology.
The term "attention-related negativity" is different from "awareness-related negativity." Attention-related negativity refers to changes in neural activity associated with selective attention to task-relevant stimuli (e.g., attention is focused on the tones presented in the left ear) versus activity elicited by the same stimuli when task-irrelevant (e.g., attention is focused on the tones presented in the right ear). The awareness-related negativity refers to differences in neural activity when participants report hearing the stimuli versus when they do not. This is now clarified.
Line 250-251: “… which aligns with the stimulation rate used in our AB experiments”. Is this related to the SOAs? “Interestingly, this 10-251 Hz entrainment was evident despite slight variations in the presentation rates across studies” Is there a more elaborate explanation for this? Why does this happen?
The variations in inter-stimulus intervals were small, with the main stimulation rate being about 10 Hz. Such a stimulation rate is expected to elicit a 10 Hz signal.
Line 256: 2x “wandering”
This is now corrected.
Line 259: Now the term awareness related negativity is used again, whereas in lines 224-226 it is stated that “The timing and amplitude distribution of the early negativity show similarities with the awareness-related negativity observed when listeners detect a pre-defined target embedded in distractors”. For me it is a bit confusing that this term is used only sometimes whereas in other parts of the manuscript the term is avoided.
We thank the reviewer for bringing this up. We have revised several sections of the manuscript to improve readability.
Line 273: why is the acronym ARN only introduced now after writing the term in full during the whole paper?
The acronym is no longer used in the discussion.